# Nurses’ Performance on the Genomic Nursing Inventory: A Cross-Sectional Study in Crete, Greece

**DOI:** 10.3390/nursrep15040121

**Published:** 2025-03-31

**Authors:** Konstantinos Giakoumidakis, Antonios Christodoulakis, Elisavet Petrogianni, Aggelos Laliotis, Alexandra Trivli, Evridiki Patelarou, Athina Patelarou

**Affiliations:** 1Department of Nursing, School of Health Sciences, Hellenic Mediterranean University, 71410 Heraklion, Greece; christodoulakisa@uoc.gr (A.C.); elisavetda@yahoo.gr (E.P.); epatelarou@hmu.gr (E.P.); apatelarou@hmu.gr (A.P.); 2Department of Social Medicine, School of Medicine, University of Crete, 70013 Heraklion, Greece; 3Department of General Surgery, Venizeleio General Hospital, Leoforos Knossou 44, 71409 Heraklion, Greece; laliotisac@gmail.com; 4Department of Ophthalmology, Konstantopouleio-Patission General Hospital, Agias Olgas 3-5, Nea Ionia, 14233 Athens, Greece; alextrivli@yahoo.com

**Keywords:** genomics, knowledge, nurses, Greece, education, translation, validation

## Abstract

**Background/Objectives:** Genomics plays a crucial role in healthcare, enhancing diagnostics, risk assessments, and therapeutic interventions. However, many healthcare professionals, including nurses, face challenges when it comes to integrating genomics into their practice. This study aims to evaluate the genomic knowledge of nurse practitioners in Greece by translating and validating the Genomic Nursing Concept Inventory (GNCI). **Methods:** A cross-sectional study involving 324 nurses was conducted in Crete, Greece. The GNCI, a 31-item questionnaire, was translated and validated for the Greek context to assess nurses’ genomics competence. **Results:** The Greek version of the GNCI demonstrated acceptable reliability (Cronbach’s alpha = 0.622). The confirmatory factor analysis indicated a satisfactory fit for the one-factor model. However, the mean GNCI score revealed significant knowledge gaps, with nurses answering only 30.1% of questions correctly. Notably, nurses showed a better understanding of genomic basics and mutations compared to inheritance and genomic healthcare applications. **Conclusions:** The present study successfully translated and validated the GNCI in Greek and highlighted important genomic-related knowledge gaps among Greek nurses, emphasizing the urgent need for more targeted educational interventions. By enhancing genomic literacy through curriculum integration and professional development, nurses could be better prepared, leading to improved patient care and healthcare outcomes in Greece.

## 1. Introduction

Genomics is a rapidly evolving field that delves into the intricate connections between all genes within an individual’s genome, elucidating their functions, and comprehending their collective impact on growth and development [1]. Using genomics, healthcare professionals can improve diagnostic times, identify at-risk individuals, and perform precision therapies that could improve health outcomes [2]. Therefore, genomics is an essential component of modern healthcare, contributing to diagnosis, risk assessment, prevention, treatment, management, and thus, overall prognosis and quality of life of patients [3]. However, despite the proven value of genomics in healthcare practice, studies have suggested that not all healthcare professionals adequately integrate genomics into their clinical practice [4,5,6,7,8]. As healthcare professionals, nurses are required to have adequate knowledge of genomics. This will enable them to better inform patients, family members, and caregivers regarding the risks of being diagnosed with various genomic-related disorders and the available treatments [9,10,11]. In addition, it is crucial for nurses to possess genomic literacy to effectively communicate with other healthcare professionals and enhance the quality of care provided to their communities [12]. However, despite the value of genomics in nursing, they are rarely incorporated into nursing education, meaning that nursing students are inadequately prepared to provide effective genetic and genomic care [13]. Hence, studies across multiple countries consistently demonstrate that nurses have insufficient knowledge of genomics [2,4,5,14,15,16]. For example, studies from the U.S. suggested moderate to low genomic knowledge in nurses [2,14,15], while Turkish nurses scored an average of 6.89 out of 11 on a genetics knowledge test [16]. This suggests that genomics is not fully incorporated into nursing curricula, as 81.1% of Turkish nurses mentioned that genetics was missing from their degree programs [16]. Similarly, gaps in knowledge among U.S. faculty indicate deficiencies in curriculum content [2,14,15]. These findings highlight significant disparities in genomic knowledge among nurses and the educational coverage across the regions studied [2,14,15,16]. Despite this, the majority of studies investigating the genomic competence of nurses have primarily relied on perceived knowledge or confidence in the content, with a few exceptions that have actually assessed actual knowledge [3,5,17,18,19,20].

Numerous validated questionnaires have been utilized over the years to assess the genomic knowledge of nurses. One such questionnaire is the Genetics Literacy Assessment Instrument, which has generated interesting data, but has limited value because it was originally designed for college science students and does not specifically measure concepts relevant to nursing [21]. To address this limitation, De Sevo (2013) developed a 15-item multiple-choice genetic/genomic knowledge test that evaluated nursing faculty performance on various topics, such as genetic definitions, inheritance patterns, referral actions, pedigree development, cultural issues, and insurance issues [22]. As a next step, Ward et al. (2014) utilized a rigorous concept inventory development strategy to create the Genomic Nursing Concept Inventory (GNCI) [23]. Concept inventories are research-based instruments designed to measure understanding of key concepts in a specific knowledge domain. The GNCI is the first concept inventory developed for nursing and aims to measure understanding of foundational genetic/genomic concepts essential to nursing practice. The concepts for the GNCI were derived from essential nursing genetic/genomic competencies [24]. This enables the GNCI to provide valuable information that allows educators to identify poorly understood concepts and address misconceptions directly. This highlights the significance of the GNCI, as it enables nurse educators and healthcare managers to comprehensively assess and enhance nurses’ genomic knowledge [23,25,26,27]. However, despite the increasing importance of genomics in nursing and the value of the GNCI, it has not been translated and validated in multiple languages, including Greek.

Nurses in Greece receive their education through a four-year Bachelor of Science degree program, with opportunities for advanced Master of Science and Doctor of Science degrees [28]. Genomics education is limited to fundamental genetics at the undergraduate level. Following formal education, there are no systematic specialized courses on genomics, although occasional workshops may be organized by professional associations, and universities [29]. Notably, no legal or regulatory framework mandates nurses to possess genomics knowledge, and such competencies are not defined for nursing roles across healthcare institutions. This highlights a substantial gap in genomic literacy among Greek nurses, indicating a broader emphasis on traditional nursing skills rather than emerging fields such as genomics. As a result, to the best of our knowledge, the genomic knowledge of nurse practitioners has not been thoroughly investigated in Greece, presenting a significant gap of empirical evidence for clinical practice. This is further emphasized by the lack of translated and validated instruments that could assess the genomic knowledge of nurses in Greek, such as the GNCI. A study that would thoroughly evaluate the genomic knowledge of these nurses could provide useful insights to improve it. Therefore, this study aimed to translate and validate the GNCI in Greek and evaluate the genomic knowledge of nurse practitioners in Crete, Greece.

## 2. Materials and Methods

### 2.1. Study Design, Period, and Setting

This cross-sectional study was conducted in four public hospitals and three primary healthcare units in Crete, Greece between July and December 2022. Specifically, a convenience sample of 408 nurses (*n* = 332 from hospitals and *n* = 76 from primary healthcare units) were invited to participate in this study. To be included, nurses had to speak Greek fluently and be actively working in one of the participating health units during the study period. Nurses who declined to participate or did not fully complete the GNCI questionnaire and the self-reported questionnaire were excluded from the study. Consequently, out of the 408 nurses who were invited, 84 did not meet the inclusion criteria (the majority of respondents who declined to participate cited a lack of time to participate and a full workload). As a result, 324 nurses were finally included in this study (response rate ~79.4%). It should be noted that approximately 2000 nurses are employed in public hospitals and primary care centers in Crete.

### 2.2. Data Collection

The nurses were approached during their shifts in hospitals or primary care centers. They were informed about the study’s objectives and then asked to complete the questionnaires at their convenience, after agreeing to participate. The nurses who agreed to participate, after signing the informed consent form, were handed (in paper format) a self-reported questionnaire that assessed their demographic characteristics (gender, age, educational level, and workplace), and the GNCI.

#### 2.2.1. The Genomic Nursing Concept Inventory (GNCI)

The GNCI was developed and validated by Ward, Haberman, and Barbosa-Leiker [23]. The GNCI consists of 31 multiple-choice items (each with one correct answer) that correspond to 18 concepts divided into four thematic groups, namely, genome basics (12 items), mutations (3 items), inheritance (8 items), and genomic healthcare (8 items). The GNCI takes 30 min to administer and has been tested with nursing students, nursing professors, and a limited number of working nurses, yielding good psychometric results [12,15,23,26,27,30,31].

#### 2.2.2. Translation and Validation of the GNCI

Two individuals individually translated the original English questionnaire into Greek. Then, a third person compared the two translations to decide on a final agreed translation (1st reconciliation version). The agreed upon version was then translated back into the language of the original questionnaire by a bilingual professional translator, who was a native English speaker but not familiar with the standard form of the questionnaire. The translated questionnaire was then completed by 10 clinical nurses from one of the participating hospitals to assess its face validity. Moreover, the nurses were asked to provide feedback and comments regarding their understanding of each GNCI item in Greek. However, they did not have any major comments or suggestions to incorporate into the Greek GNCI. This confirmed that the questions in the scale were consistent with the attribute being measured and did not lead to incomplete or misleading responses [32]. These 10 clinical nurses were only involved in the translation process and did not participate in any other parts of the study. They were invited to participate in June 2022, following the aforementioned inclusion and exclusion criteria.

### 2.3. Data Analysis

Each of the 31 GNCI questions was scored as correct (1 point) or incorrect (0 points), and a total score ranging from 0 to 31 was calculated. Afterward, the total score was converted to a percentage score for each participant [15]. Subsequently, percentage scores for the four topical categories were also calculated by summing the number of correct answers for the specified items in each category. Internal consistency was assessed using Cronbach’s alpha. A Cronbach’s alpha coefficient greater than 0.6 indicates acceptable reliability for research purposes and suggests that the items are interdependent and homogeneous in measuring the construct [33]. To assess the fit of the model, Confirmatory Factor Analysis (CFA) was performed on the participating population of 324 clinical nurses. A Standardized Root Mean Squared Residual (SRMR) less than or equal to 0.08, a Coefficient of determination (CD) greater than or equal to 0.90, and a Comparative Fit Index (CFI) greater than or equal to 0.90 indicates an adequate or good fit. Continuous variables were presented as mean ± standard deviation (SD), while categorical variables were expressed as numbers (percentages). The statistical analyses were conducted using STATA (version 12) for the CFA and IBM SPSS (version 26) for the remaining analyses. Statistical significance was set at an alpha level of 0.05.

## 3. Results

Most participants (80.9%) were female, with an average age range of 23 to 59 years (Table 1). In terms of their educational level, the majority of participants held a bachelor’s degree (80.6%), and most of them were employed at a hospital (81.8%).

Cronbach’s alpha for the 31 items of the translated version of the GNCI was 0.622. The contribution of each item to the scale can be examined in (Appendix A). However, a one-factor model was conducted by CFA (Figure 1), giving acceptable global fit indices. The resulting global fit indices (SRMR = 0.076, CD = 0.677, CFI = 0.744) suggested that the 31 items in the one-factor solution proposed by the primary researchers could be accepted for the Greek GNCI.

Overall, participants answered only 30.1% of all GNCI questions correctly, indicating significant gaps in genomic knowledge (Table 2). Participants achieved a mean total score of 9.34 out of 31 possible points (SD = 3.90, 95% CI: 8.91–9.77). Moreover, participants had a slightly better understanding of basic genome and mutation concepts than inheritance patterns and the applications of genomics in clinical practice (Table 2).

The GNCI item-by-item examination indicated considerable differences in participants’ understanding of individual genomic concepts (Table 3). More specifically, the highest-scoring items were Genome Organization/item 8 (“Rank the following genetic structures in terms of size starting with the largest and proceeding to the smallest: chromosome, gene, genome, nucleotide”) (highest accuracy 52.5% correct), followed by Autosomal Recessive Inheritance/item 16 (50.0% correct), indicating a moderate understanding of basic genome structure and this specific inheritance pattern (parent to child). On the other hand, some of the lowest-scoring items revealed critical knowledge gaps. For example, X-linked Inheritance/item 17 had the lowest accuracy in the entire inventory (13.3% correct), indicating a more substantial lack of understanding of how X-linked traits are inherited. Similarly, Genome Organization/item 4 (“The human insulin gene has been identified and designated INS. Which cells in your body contain the insulin gene?”) and Heterozygosity/item 13 both had a very low percentage of correct answers (16.4%), suggesting a significant gap of knowledge related to genome structure and variation. Interestingly, the Genome Organization/item 8 (“Rank the following genetic structures in terms of size, starting with the largest and proceeding to the smallest: chromosome, gene, genome, nucleotide”) and Genome Organization/item 4 (“The human insulin gene has been identified and designated INS. Which cells in your body contain the insulin gene?”) were scored as the highest and lowest, respectively. This suggests that nurses possess a basic understanding of general genomic concepts (i.e., size of chromosomes, genes, genome, and nucleotides), but lack specific knowledge (i.e., cells containing the insulin gene). More importantly, participants scored exceptionally low in items that examined their capacity to apply genomic knowledge in clinical scenarios. For example, Carrier Testing/item 22 received only 16.0% correct responses, suggesting a limited understanding of how genetic screening is utilized in clinical care. Similarly, Benefit of Family Health History/item 26 had only 20.1% correct responses, indicating that participants may not fully grasp the clinical importance of family history when assessing genomic risks. Meanwhile, participants performed somewhat better on Pharmacogenomics/item 28 (38.3% correct), suggesting some familiarity with the role genetics plays in drug response. Nevertheless, the overall pattern of responses suggests that participants lacked a comprehensive understanding of genomics and that applying these concepts in clinical decision-making poses a significant challenge for them.

## 4. Discussion

The main objective of this study was to translate and validate the GNCI in the Greek language. Additionally, the study aimed to evaluate the genomic knowledge of nurses working in Crete, Greece. The internal consistency of the Greek version of the GNCI was found to be acceptable, with a Cronbach’s alpha of 0.622. Moreover, our findings suggest that Greek nurses have significant gaps in genomic knowledge (overall 30.1% of correct answers), with performance varying across different categories (Genome Basics, Mutations, Inheritance, and Genomic Health Care). Furthermore, we also found that nurses have substantial knowledge deficiencies in the integration of genomics into their clinical practice.

The present study has successfully translated and validated the GNCI into Greek, thus establishing a valuable tool for assessing and improving the genomic knowledge of Greek nurses. Although the internal consistency was acceptable (Cronbach’s alpha = 0.622), this value is slightly below the commonly accepted threshold of 0.70 [33]. The developer of the GNCI reported a Cronbach’s alpha of 0.77 [23]. In addition, another study found that the Cronbach’s alpha of the GNCI was 0.78 [25]. Nevertheless, a Cronbach’s alpha higher than 0.6 is considered adequate for exploratory research and initial validation studies, such as the present study [34]. This is particularly true for the Greek GNCI, since CFA (SRMR = 0.076, CD = 0.677, CFI = 0.744) further supported its reliability and validity. Previous research has also yielded comparable CFA results [26,30,35,36,37,38], which suggests that the items on the Greek GNCI are interconnected, and collectively represent a unified construct for genomic knowledge in nurses [26,39,40]. Furthermore, the CFA results indicate that the Greek translation of the GNCI preserves the structural integrity of the original instrument, thereby confirming its suitability for use in the Greek context [26,39,40]. Consequently, our study has successfully translated and validated the GNCI in Greek, although future studies could further refine each item and improve the Greek GNCI’s Cronbach’s alpha.

An important finding of the present study was that the nurses who participated in the study had a low level of genomic knowledge. Other studies align with this finding and confirm that nurses have low levels of genomic knowledge [12,17,37]. For example, a cross-sectional study in Jordan with 751 nurses found even lower mean scores than in our study [12]. Another cross-sectional study conducted with 253 Australian nurses also found low levels of genomic knowledge in their participants [17]. A possible explanation for this finding is that nurses face numerous barriers that restrict them from improving their genomic knowledge, such as time constraints and unawareness of available educational materials [37]. On the other hand, other studies suggest that nurses have low to moderate levels of genomic knowledge [5,15,18,36,41,42]. An integrative review revealed that nurses frequently lack a deep understanding of crucial areas, such as genetic inheritance and pharmacogenomics, which impairs their ability to fully integrate genomics into patient care [5]. A cross-sectional study in the U.S. utilizing the GNCI further demonstrated that nurses possess only low to moderate levels of genomic literacy, particularly struggling with interpreting genetic test results and applying genomic principles clinically, aligning with our study [15]. Similarly, a mixed-method systematic review highlighted that inadequate training leaves nurses unprepared to discuss genomic information with patients [18]. Meanwhile, a nationwide survey in China suggested that while nurses possess moderate knowledge, they encounter challenges with advanced genomic applications, compounded by barriers such as time constraints and insufficient institutional support [36]. A study in Australia has further suggested that nursing curricula often provide only basic genetics education, with comprehensive genomics coverage and post-education training remaining inconsistent and inadequate [41]. On the other hand, a study in Taiwan emphasized that nurses themselves had a strong desire for more training and clearer guidelines, and they acknowledged the critical role of genomics in contemporary healthcare [42]. These findings underscore the urgent need for educational reform, structured continuing education programs, and global collaboration among nursing organizations to enhance genomic literacy and ultimately improve patient care outcomes. Global collaboration between nurse professional organizations also presents a potential and perhaps necessary solution to the lack of genomic knowledge among nurses worldwide.

Our findings also suggested that nurses lack the necessary knowledge to adequately integrate genomics into their clinical practice. This is in line with previous research that suggests healthcare professionals, including nurses, often have limited knowledge of genomics, which can potentially impact patient care and clinical decision-making [6,43]. For example, a study conducted by Calzone et al. (2018) highlighted similar trends, showing that nurses around the world face challenges in incorporating genomics into their practice due to insufficient knowledge and training [6]. Additionally, the lack of understanding in areas such as X-linked inheritance and carrier testing indicates specific gaps in knowledge that align with the findings of Costa et al. (2024), who emphasized the importance of targeted educational programs to improve their understanding of complex genetic concepts [43]. These deficiencies highlight the need for educational reforms and the inclusion of genomics in nursing curricula to better equip healthcare professionals with the necessary skills to effectively apply genomic information in patient care [44]. Moreover, the discrepancy in understanding specific genomic concepts, such as a higher level of proficiency in basic genome structure compared to clinical application items, like carrier testing, reflects a broader issue in healthcare education. The literature consistently shows that theoretical knowledge often exceeds practical application, as noted by Campion et al. (2019), who reported similar findings in their assessment of genomic education among healthcare providers [45]. This suggests that, although nurses may understand fundamental genomic concepts, they struggle to apply this knowledge in real-world clinical scenarios, particularly in patient counseling and risk assessment. Therefore, there is an urgent need to address these gaps through including enhancing education and training programs to boost genomic literacy among nurses, integrating cultural competence and implicit bias awareness into genomics education, and incorporating genomic content into undergraduate and continuing education curricula [45]. On the other hand, professional nursing organizations could develop and utilize initiatives that aim to advance genomics education and advocacy within nursing communities [46].

In Greece, nurses’ level of involvement in genomics is minimal and indirect, according to our study. Typically, nurses are not the primary decision-makers when it comes to ordering or interpreting genomic tests, as these roles are usually reserved for physicians or geneticists [28]. However, nurses can still play supportive roles, such as providing patient education, collecting samples for genetic testing, and coordinating care following genomic-based diagnoses or treatments [46]. Although there are resources for genomics education in Greece, they are limited and mainly targeted towards physicians and researchers rather than a wider range of healthcare professionals, including nurses. On the other hand, academic institutions offer genetics and genomics courses within medical, biological, or biomedical science undergraduate programs [47]. However, formal genomics education for nurses in undergraduate nursing programs in Greece is scarce, with only a few courses covering aspects of genomics, like biology and genetics [29]. Opportunities for continuing professional development, such as workshops or short courses, may be available through professional associations or research institutions, but they are not widely accessible or systematically organized [29]. Occasionally, training may be offered through international collaborations or European Union-funded projects, but these are not specifically tailored to Greek nurses [48]. Furthermore, apart from the lack of education, the lack of genomic knowledge among Greek nurses may be attributed to various reasons, such as limited practical experience with genomic applications and the absence of formal guidelines on genomic knowledge for nurses in Greece [49]. Nurses may also perceive genomics as a field that is exclusive to physicians or geneticists, which could reduce their motivation to learn about it, contributing to the knowledge gaps in genomics [49].

Given the significant knowledge gaps identified by our study, we suggest that, to address both theoretical and practical aspects of genomics in nursing, it is essential to implement educational interventions in nursing education. For example, the incorporation of genomics into both undergraduate Greek nursing curricula and continuing professional development programs for nurses and nurse educators [22,31,50]. This could enable nurses to better apply genomic information in clinical settings, thus improving patient care and outcomes [3,6,51,52,53]. These educational interventions could include workshops, online courses, and real-world case studies that emphasize the practical application of genomic knowledge in clinical settings [6,45]. Another potential strategy to enhance nurses’ comprehension of genomics entails healthcare managers and nurse educators collaborating to utilize reliable tools such as the GNCI for assessing and augmenting nurses’ knowledge in this domain [2,23]. Subsequently, targeted interventions can be integrated to further improve their understanding [45]. Integrating comprehensive genomic education into nursing programs could better equip nurses for the changing landscape of healthcare, where genetic information plays an increasingly important role in clinical decision-making [2,45,54]. By improving genomic literacy, nurses could provide more informed patient care, actively contribute to multidisciplinary teams, and contribute to the overall improvement of healthcare delivery in the era of genomics [55,56].

### Limitations

This study, which identified significant gaps in genomic knowledge among Greek nurses, successfully translated and validated the GNCI in Greek. However, despite its advantages, it is important to acknowledge a few limitations that warrant attention. First, the sample size was limited to nurses in Crete, which may not accurately represent nurses from all regions of Greece. Therefore, future research could include a more diverse sample and utilize longitudinal designs to track changes in genomic knowledge over time. Second, the moderate internal consistency of the GNCI, as shown by Cronbach’s alpha. Although, as stated, this consistency is acceptable for research purposes, future studies should aim to improve the instrument’s reliability by refining individual items or considering additional items that better capture the construct of genomic nursing knowledge. Third, the significant disparity in the allocation of nurses between hospitals and primary care settings may have impacted the outcomes, as there could be variations in experience and exposure to genomic knowledge in these two environments. Hence, future research should also consider this factor, when designing studies that assess Genomic knowledge in nurses. Finally, participants required 30 min to finish the GNCI, which could have led to fatigue [57]. Hence, in future studies, it would be beneficial to further adjust and minimize the size of GNCI to enhance completion time.

## 5. Conclusions

In conclusion, our study successfully translated and validated the GNCI into Greek. Moreover, our findings revealed a concerning lack of genomic knowledge among nurses in Crete, Greece. This emphasizes the urgent need for systemic reforms in nursing education, focusing on both fundamental genomic concepts and their practical implementation in clinical settings. Considering the rapid integration of genomics in modern healthcare, it is crucial to enhance nurses’ genomic competencies to improve patient care outcomes. Meanwhile, the GNCI could offer a valuable framework for educators and healthcare managers to design customized educational strategies, ensuring that nurses are well-prepared to integrate genomic insights into their practice. Future studies could encompass a wider range of geographical regions in Greece, providing a more comprehensive understanding of genomic literacy of nurses across diverse healthcare environments. Additionally, refining the GNCI items to enhance internal consistency could further validate its applicability in broader contexts. Nevertheless, this study sets the foundation for a future where genomic knowledge becomes a fundamental aspect of nursing practice, ultimately leading to improved patient care and outcomes in the era of precision health and evidence-based practice.

## Figures and Tables

**Figure 1 nursrep-15-00121-f001:**
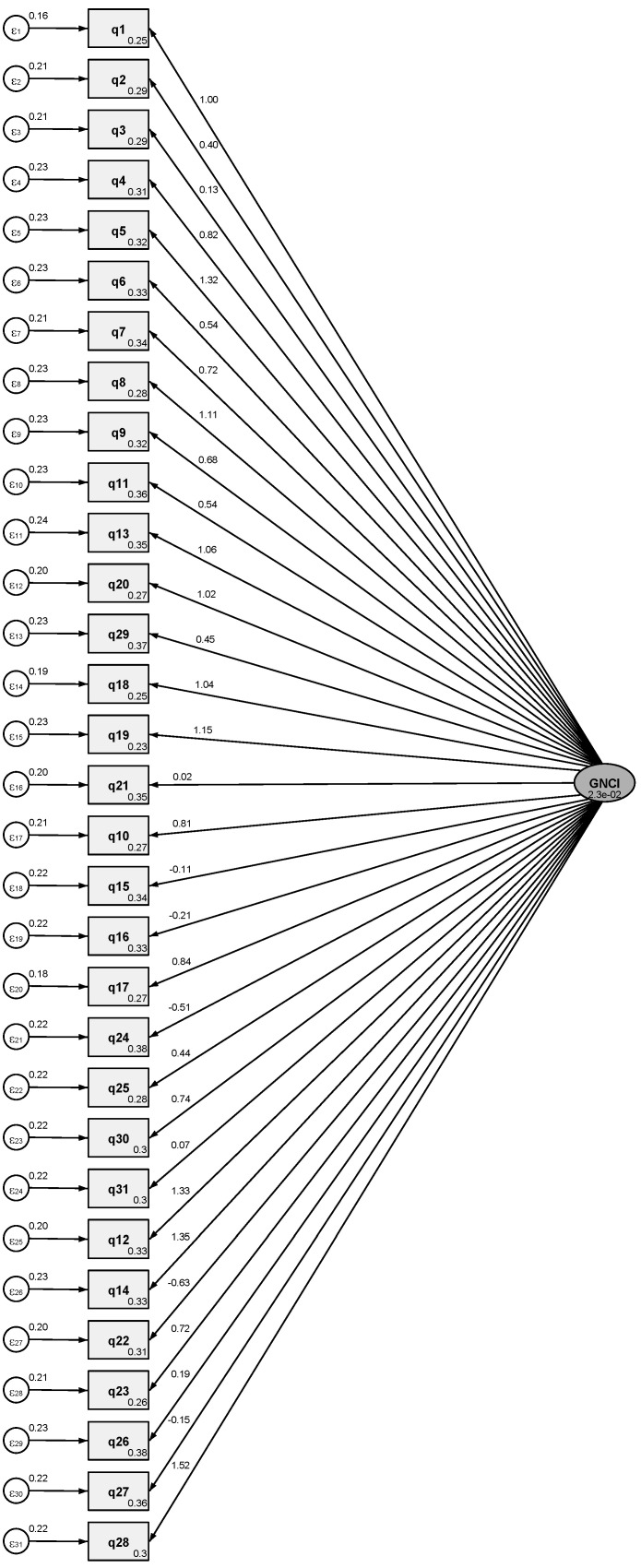
Confirmatory factor analysis of the Greek Genomic Nursing Concept Inventory (GNCI).

**Table 1 nursrep-15-00121-t001:** Demographic characteristics of the 324 participating nurses.

**Age**	Mean (SD)		41.0 (8.2)
**Biological sex**	Female	Ν (%)	262 (80.9%)
	Male	Ν (%)	62 (19.1%)
**Educational level**	BSc	Ν (%)	261 (80.6%)
	MSc/PhD	Ν (%)	63 (19.4%)
**Workplace**	Hospital	Ν (%)	265 (81.8%)
	Primary healthcare unit	Ν (%)	59 (18.2%)

**Table 2 nursrep-15-00121-t002:** Total score and subscales performance (correct answers) in the GNCI of the 324 participating nurses.

Category	Mean Score (SD)	95% CI (Lower–Upper)	% Correct
Total Score (31 Items)	9.34 (3.90)	8.91–9.77	30.1%
Genome Basics (12 Items)	3.78 (1.98)	3.56–4.00	31.5%
Mutations (3 Items)	0.99 (0.80)	0.90–1.07	33.0%
Inheritance (8 Items)	2.31 (1.63)	2.13–2.49	28.9%
Genomic Health Care (8 Items)	2.26 (1.65)	2.08–2.44	28.3%

**Table 3 nursrep-15-00121-t003:** Total number and percentage of correct responses for each GNCI Item of the 324 participating nurses.

Item #	Ν	% of Total Correct Answers
Item 1: Gene function	131	40.4 %
Item 2: Genome organization	166	51.2%
Item 3: Human genome homogeneity	70	21.6%
Item 4: Genome organization	53	16.4 %
Item 5: Genome composition	59	18.2 %
Item 6: Gene function	82	25.3 %
Item 7: Genotype-phenotype association	93	28.7 %
Item 8: Genome organization	170	52.5 %
Item 9: Gene function	131	40.4 %
Item 10: Dominance	72	22.2%
Item 11: Gene expression	116	35.8%
Item 12: Pharmacogenomics	105	32.4 %
Item 13: Heterozygosity	53	16.4 %
Item 14: Genetic screening tests	117	36.1 %
Item 15: Autosomal recessive inheritance	91	28.1 %
Item 16: Autosomal recessive inheritance	162	50.0 %
Item 17: X-linked inheritance	43	13.3 %
Item 18: Germline/somatic mutations	116	35.8 %
Item 19: Mutation heterogeneity	96	29.6 %
Item 20: Cancer genotyping	106	32.7 %
Item 21: How mutations cause disease	108	33.3 %
Item 22: Carrier testing	52	16.0 %
Item 23: Family history—red flags	77	23.8 %
Item 24: Inheritance of autosomal mutations	89	27.5 %
Item 25: Genetics of multifactorial conditions	83	25.6 %
Item 26: Benefit of family health history	65	20.1 %
Item 27: Pharmacogenomics	87	26.9 %
Item 28: Pharmacogenomics	124	38.3 %
Item 29: Heterozygosity in autosomal dominant conditions	101	31.2 %
Item 30: Autosomal dominant inheritance	90	27.8 %
Item 31: Autosomal dominant inheritance	118	36.4 %

## Data Availability

The data that support the findings of this study are available from the corresponding author upon reasonable request.

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
