# Peer review of "Nurses’ Performance on the Genomic Nursing Inventory: A Cross-Sectional Study in Crete, Greece"

_nursrep, 2025, doi:10.3390/nursrep15040121_

Round 1

Reviewer 1 Report

Comments and Suggestions for Authors

Thank you for the opportunity to review this manuscript. The application of genomics is rapidly expanding and nurses play a significant role in optimising this therefore, assessing baseline knowledge to enable appropriate education interventions is key. Overall the article is well-written and a sound methodology was applied. I have a few comments for the authors to consider:

  1. Would it be more appropriate to include Crete in the title as the area where the study is conducted. The title is slightly misleading when it refers to the whole country of Greece?
  2. Comments on the methods:
    • How many nurses are there in Crete to establish the proportion of nurses studied vs the target population?
    • Did the 10 nurses who initially trialled the questionnaire or participants have any feedback or seek any clarification on the translated versions?
    • Where there any reasons given by the 84 nurses who did not finish the questionnaire?
    • What was the setting of of the administration of the questionnaire? Were they timed altogether in a room for 30 mins or was this just handed out in the workplace whilst they were on shift etc.
  3. Discussion-I would like to know more about the context of genomics in Greece ie. is there a national programme funded by the government? How is it implemented in general and what is the extent of nurse involvement? Are there resources existing in Greece for genomics education (for any profession)? What could be other reasons apart from lack of education that resulted in low scores? What about the conduct of the study solely in Crete-does the setting have any impact on the results?

Author Response

Dear esteemed reviewer,

I would like to thank you for your time and effort, and I hope that the answers meet your expectations.

Reviewer 2 Report

Comments and Suggestions for Authors

Dear authors,

The manuscript's topic is very current, and in addition to the confirmed questionnaire validation, it indicates necessary educational interventions to improve patient care and health outcomes in the region/country where the research was conducted. The study uses a robust methodology that is rigorously explained.

I would like to make some suggestions that could improve the manuscript:

Introduction

The introduction systematically presents the framework of the problem addressed by the manuscript, highlighting the many aspects of genomics that contribute to quality health care.

For a better understanding of the importance of the topic of the manuscript, I suggest that after the general statements in lines 51-52, specific data should be provided on which professions in the region/country were found to have insufficient knowledge of genomics; data from the references in line 52 would be sufficient. It would be particularly important to provide what data were obtained on nurses' knowledge, in which countries, and whether and to what extent aspects of genomics are studied in their level of education/curriculum.

In the final part of the Introduction section, before the paragraph starting in line 75, for a better understanding of the needs of conducting the study, it is significant whether and to what extent genomics is studied during the formal education of various health professionals in the country where the study was conducted. In particular, you should describe how many levels of training nurses have, whether and how much they learn about genomics, whether there are special courses after formal education, and who organizes them. Also, whether in practice or different types/levels of health care institutions, nursing professional competencies that require knowledge of genomics are defined by rules/law. Please provide more information.

Methods

Data on the study's design, participants, data collection, instruments, translation and validation are precisely presented.

Results

The results are presented in tables with adequate accompanying text.

Discussion

The discussion is extensive, and the authors draw attention to many important questions raised by their research and the results of other authors.

Are the statements in lines 260 to 264 referring to Greek nurses? References should support the statements in lines 265 to 276. I suggest you correct/complete them.

The conclusions are concise, well-argued, and based on research results. The references mentioned are relevant to the topic that the paper dealt with.

I hope you find my comments helpful.

Author Response

(The authors gave the same response as above.)

Reviewer 3 Report

Comments and Suggestions for Authors

Thank you for reporting on an interesting study that expands the global database about nurses’ genomic knowledge. Please see my suggested edits below:

Title: Suggest avoiding the term “nurse practitioner” as this has a specific meaning in North America and elsewhere as a nurse with additional education beyond a baccalaureate degree which does not seem to the population studied.

Abstract/Key Words: suggest using MeSH terms for key words

Introduction:

Line 41 and line 49, suggest replacing “and despite” with just “despite”

Line 51: suggest replacing reference with a more current reference, the new Essentials was published in 2023

Line 55: please add citations

Line 70 and line 80: please change to “the GNCI”

Line 82: delete “and” in “genomic knowledge and of nurse practitioners…”

Materials and Methods:

Line 100: “The GNCI…”

Line 100: What is meany by “weighted”?

Line 122: Separate into 2 sentences

Results:

Why was a McNemar test used if there was no pretest/posttest? Or was there and this was not explained in the methods?

Lines 158 – 164 are repeated from Table, so could be deleted

Line 175: May want to add a bit more on the difference in questions about genome organization. Why was one higher and the other lower?

Discussion:

Lines 205 and 206 and 215: “The GNCI..please check for this throughout manuscript

Suggest expanding section comparing your GNCI findings with recent studies from additional countries and being more specific with the comparisons. Also think about the possibility and perhaps necessity of global collaboration on the issue of low genomic knowledge among nurses.

Line 234: I would say that you found a suggestion of this, but not directly that there is a lack of integration into practice.

Line 257: this is an old reference. What are more current strategies that are being used to address gaps in nursing knowledge of genomics?

Line 310-311: suggest using “precision health” instead of personalized medicine

Look for some repetition throughout discussion.

Author Response

(The authors gave the same response as above.)
